# Brain network coupling associated with cognitive performance varies as a function of a child's environment in the ABCD study

Monica E. Ellwood-Lowe [1✉], Susan Whitfield-Gabrieli[2] & Silvia A. Bunge [1,3]

Prior research indicates that lower resting-state functional coupling between two brain networks, lateral frontoparietal network (LFPN) and default mode network (DMN), relates to cognitive test performance, for children and adults. However, most of the research that led to this conclusion has been conducted with non-representative samples of individuals from higher-income backgrounds, and so further studies including participants from a broader range of socioeconomic backgrounds are required. Here, in a pre-registered study, we analyzed resting-state fMRI from 6839 children ages 9–10 years from the ABCD dataset. For children from households defined as being above poverty (family of 4 with income > $25,000, or family of 5+ with income > $35,000), we replicated prior findings; that is, we found that better performance on cognitive tests correlated with weaker LFPN-DMN coupling. For children from households defined as being in poverty, the direction of association was reversed, on average: better performance was instead directionally related to stronger LFPN-DMN connectivity, though there was considerable variability. Among children in households below poverty, the direction of this association was predicted in part by features of their environments, such as school type and parent-reported neighborhood safety. These results highlight the importance of including representative samples in studies of child cognitive development.

[1] Department of Psychology, University of California, Berkeley, Berkeley, CA, USA. [2] Department of Psychology, Northeastern University, Boston, MA, USA. [3] Helen Wills Neuroscience Institute, University of California, Berkeley, Berkeley, CA, USA. ✉email: mellwoodlowe@berkeley.edu

In the United States, one-fifth of children are estimated to live below the poverty line (defined based on whether their income is high enough to meet their basic needs, for a household of their size[1]). Relative to children living just above poverty, these children are least likely to have access to the federal social safety net (e.g., government-provided medical care, tax credits, or food supplements), and they are at heightened risk for poor health and educational outcomes[2,3]. Compared to their peers whose families have higher incomes, children living in poverty tend to perform worse on tests of cognitive functioning[4]. However, such broad comparisons obscure substantial variability within the group of children from households in poverty, a large segment of whom score on par with their higher-income peers on canonical cognitive tests. Here, we seek to understand neural correlates of high cognitive test performance in children from households in poverty. We examine the neural and environmental correlates of cognitive test performance in a sample of over 1000 children across the United States estimated to be living in poverty based on their household size and income.

Over the past decade, researchers have documented neural differences between children from higher and lower SES backgrounds in brain structure and function from an early age[5–7]. However, even within this literature, children from households living below the poverty line tend to be underrepresented in studies. In addition, many studies compare children from higher and lower SES backgrounds, and do not address variability within the lower SES group. Thus, characterizing brain development associated with high test performance in children from households living below poverty could help shift our questions away from how these children differ from children from households above poverty, and toward understanding mechanisms supporting neurocognitive performance across the spectrum of human experience. This is particularly important given that children living in poverty can have different experiences from those who are typically studied in developmental cognitive neuroscience[4,8,9].

Accumulating evidence suggests that the brain adapts to the affordances and constraints of an individual's environment, especially in early life. Indeed, a growing number of studies have complicated the notion that there is an ideal childhood environment, suggesting that different environments promote the development of distinct, adaptive cognitive skills[10–12]. Additionally or alternatively, individuals growing up experiencing different external pressures may develop the same level of cognitive proficiency but do so via different neural mechanisms[13].

In line with the hypothesis that children may achieve the same behavioral outcome through different developmental routes, studies examining brain function during higher-level cognitive tasks have found qualitatively different brain-behavior relations as a function of children's family income. In particular, children from households with lower versus higher household incomes may differentially engage higher-order brain areas such as lateral prefrontal and parietal regions to complete tasks that use working memory, rule learning, and attention[14–16]. These differences in task-related brain activation are typically thought to reflect differences in either the cognitive mechanisms by which children approach the task or the efficiency of neural processing.

Another approach for understanding environmental influences on brain development is to measure brain function in the absence of specific task demands: measuring slow-wave fluctuations in neural activity over time while participants lie awake in an MRI scanner. This approach, called resting-state fMRI, has revealed temporal coupling—so-called functional connectivity—among anatomically distal brain regions that form large-scale brain networks[17]. Cognitive networks typically become more cohesive and segregated from one another across development[18,19]. Patterns of temporal coupling within and across resting-state networks reflect regions' prior history of co-activation, offering insight into individuals' recent thought pattern[20]. Thus, resting-state fMRI can be leveraged to assess how everyday experience shapes brain networks.

Several large-scale brain networks have been linked to higher-level cognition. In particular, the lateral frontoparietal network (LFPN) is consistently activated in higher-level cognitive tasks, such as those using executive functions or reasoning[21]. Regions in the LFPN are more active during the performance of cognitively demanding tasks than during rest periods[22]. In contrast, regions in the default mode network (DMN), including regions in the medial frontal and medial parietal areas, are consistently deactivated during focused task performance. These regions have been implicated in unconstrained, internally directed thought[23], as well as during the performance of tasks that require introspection, mentalizing about others, or other mentation outside of the here-and-now[24].

Thus, LFPN and DMN have often been characterized as opponent networks. Elevated DMN activation during attention-demanding tasks has been associated with lower and more inconsistent cognitive test performance among adults[25–27]. Similarly, weaker resting-state connectivity between LFPN and DMN, and stronger connectivity among LFPN regions, have been associated with better cognitive test performance[26,28–30]. Together, these findings suggest that, in order to complete a cognitively demanding task, individuals must focus narrowly on the task at hand while inhibiting internally-directed or self-referential thoughts[23,27,31,32].

This conclusion that separation between LFPN and DMN is better for cognitive performance has been bolstered by fMRI research in typically developing children, both in terms of age-related changes and individual differences. First, there is evidence that the LFPN and DMN functionally segregate during childhood. Key nodes in the LFPN and DMN appear positively correlated in middle childhood, anti-correlated in adolescence, and more strongly anti-correlated during young adulthood[33]. Further, as with adults, children ages 10–13 who showed less coupling than their same-age peers tended to have higher cognitive test scores[34]. Tighter coupling between key nodes in these networks at age 7 has even been shown to predict increased attentional problems over the subsequent four years[35]. The conclusion drawn from these studies is that it is adaptive for LFPN and DMN to become decoupled—or even negatively coupled—during the performance of a cognitively challenging task and that the development of this dissociation may promote a stronger focus on externally directed tasks.

Despite this coherent body of findings regarding LFPN and DMN and their interactions, several points bear mentioning. First, there is evidence that LFPN and DMN interact during the performance of tasks that benefit from internally directed cognition, or mentation outside of the here-and-now[24,36–38]. Second, because the vast majority of fMRI studies involve samples with participants from relatively high SES backgrounds, we do not know whether the reported brain-behavior relations are universal. Indeed, there is evidence that children and adolescents living in socioeconomically disadvantaged neighborhoods show differences in resting-state connectivity patterns, some of which correlate with anxiety symptomatology[39]. Further, changes in family income in adolescence have been associated with changes in DMN connectivity[40]. It is important to understand both how these differences arise, and whether or how they are behaviorally relevant.

Drawing from a large behavioral and brain imaging dataset including over 10,000 children across the United States (ABCD Study[41]), we asked whether the patterns of connectivity that are adaptive among children from higher-SES backgrounds are also

associated with high cognitive test performance in children from households living in poverty. Specifically, in a set of pre-registered analyses, we tested whether characteristics of LFPN and DMN connectivity were associated with cognitive test performance for over 1000 children from this larger dataset who were estimated to be living in poverty. These children had a total family income below $35,000 (below $25,000 for children in families of 4 or less[42]), which differs from the sample composition of most prior studies in terms of SES. We sought to assess children's performance on higher-level cognitive tasks that did not task verbal skills, given well-established SES differences in verbal performance[16]. Thus, we combined measures of children's abstract reasoning (Matrix reasoning task), inhibitory control (Flanker task), and cognitive flexibility (Dimensional Change Card Sort task). This dataset had a tight age range, from 9 to10 years, minimizing variance related to age differences.

Given prior findings based largely on higher-SES children and adults, we predicted that weaker LFPN-DMN between-network connectivity and stronger within-network LFPN (i.e., LFPN-LFPN) connectivity would be related to higher cognitive test performance in children living in households in poverty. Alternatively, however, these children might show different brain-behavior associations in comparison to those seen in the prior literature, given different early experiences. In line with theories that children could develop the same cognitive outcomes through alternate developmental trajectories, one might expect that higher cognitive test scores would be associated with different patterns of network connectivity among children from households in poverty compared to findings from children from higher SES backgrounds in previous studies. Indeed, our analyses revealed a different pattern in children in households in poverty than had been observed in prior studies of children from higher SES backgrounds. As a result, we conducted follow-up analyses involving children from households with higher incomes in this sample to test whether this analysis would replicate prior findings and confirmed that it did.

In the second set of pre-registered analyses, we analyzed demographic variables to better understand features of children's environments which might explain variability both in their cognitive test performance and in the relation between LFPN-DMN connectivity and cognitive test performance. We looked at a set of 29 variables that encompass home, school, and neighborhood contexts to see whether they could predict variability in test performance in children living in poverty in this sample. We also included interactions between LFPN-DMN connectivity and each of these variables, to see if patterns of brain-behavior relations could be explained by any particular set of environmental variables.

## Results

We identified 1,034 children between ages 9 and 10 with usable data on cognitive test performance, resting-state fMRI, and demographic characteristics, whose households were likely to be below the poverty line at the time the data were collected (2016–2018), based on their self-reported household income bracket and their household size. We identified an additional 5,805 children from the same study sites who had usable data on the same measures and whose households were likely to be *above* the poverty line. Participant information is displayed in Table 1 and Supplementary Table 1.

Children's scores on the three cognitive tests (Matrix reasoning, Flanker task, and Dimensional Change Card Sort task) were moderately correlated with each other, $r = 0.23$–$0.43$ in the whole sample, $r = 0.25$–$0.39$ for the below-poverty group. We created summary cognitive test scores by summing children's

**Table 1 Participant characteristics.**

| | Above poverty (n = 5805) | Below poverty (n = 1034) | p-test |
|---|---|---|---|
| Age in months (mean (SD)) | 119.44 (7.54) | 118.89 (7.50) | 0.032 |
| Sex at birth (%) | | | 0.055 |
| Other/did not disclose | 0 (0.0) | 1 (0.1) | |
| Female | 2913 (50.2) | 511 (49.4) | |
| Male | 2892 (49.8) | 522 (50.5) | |
| Primary caregiver in study (%) | | | <0.001 |
| Biological mother | 4904 (84.5) | 920 (89.0) | |
| Biological father | 645 (11.1) | 54 (5.2) | |
| Adoptive parent | 137 (2.4) | 18 (1.7) | |
| Custodial parent | 43 (0.7) | 23 (2.2) | |
| Other | 76 (1.3) | 19 (1.8) | |
| Site (de-identified) (%) | | | <0.001 |
| site02 | 429 (7.4) | 19 (1.8) | |
| site03 | 285 (4.9) | 130 (12.6) | |
| site04 | 369 (6.4) | 122 (11.8) | |
| site05 | 203 (3.5) | 42 (4.1) | |
| site06 | 395 (6.8) | 16 (1.5) | |
| site07 | 170 (2.9) | 42 (4.1) | |
| site08 | 177 (3.0) | 14 (1.4) | |
| site09 | 250 (4.3) | 24 (2.3) | |
| site10 | 297 (5.1) | 101 (9.8) | |
| site11 | 224 (3.9) | 67 (6.5) | |
| site12 | 298 (5.1) | 73 (7.1) | |
| site13 | 361 (6.2) | 61 (5.9) | |
| site14 | 434 (7.5) | 15 (1.5) | |
| site15 | 127 (2.2) | 85 (8.2) | |
| site16 | 820 (14.1) | 70 (6.8) | |
| site18 | 208 (3.6) | 19 (1.8) | |
| site20 | 422 (7.3) | 76 (7.4) | |
| site21 | 314 (5.4) | 54 (5.2) | |
| site22 | 22 (0.4) | 4 (0.4) | |
| *RSfMRI mean framewise displacement (mean (SD))* | *0.19 (0.15)* | *0.23 (0.18)* | *<0.001* |
| *LFPN-DMN connectivity (mean (SD))* | *0.058 (0.06)* | *0.061 (0.06)* | *0.061* |
| *LFPN-LFPN connectivity (mean (SD))* | *0.21 (0.07)* | *0.21 (0.08)* | *0.286* |
| *Matrix reasoning raw score (mean (SD))* | *18.67 (3.51)* | *16.35 (3.89)* | *<0.001* |
| *Flanker raw score (mean (SD))* | *95.34 (8.03)* | *91.92 (10.24)* | *<0.001* |
| *Card sort raw score (mean (SD))* | *94.09 (8.58)* | *89.83 (9.79)* | *<0.001* |

Plain text: Demographic information; *italics*: Brain and cognitive variables.
Demographic information in plain text; brain and cognitive variables italicized. *LFPN* Lateral frontoparietal network. *DMN* Default mode network. *RSfMRI* Resting state functional magnetic resonance imaging. *P*-values without correction obtained from two-sided t-tests, calculated using the tableone package in R.

standardized scores on all three tests, as pre-registered. We first tested whether there was an association between income and cognitive test scores, using a linear mixed-effects model with a random intercept for the study site. For the purposes of comparison to prior studies, income was operationalized (for this analysis only) as a pseudo-continuous variable, using the median income level in each income bracket. Results replicated prior studies (e.g.,[43–45]): on average, children whose families had higher incomes tended to perform better on cognitive tests, $B = 0.008$, $SE = 0.0004$, $p < 0.001$, $r = 0.24$, a moderate effect size, though it accounts for only 6% of the variance in children's cognitive test scores. As shown in Fig. 1, however, there was large individual variability in cognitive test scores within each income bracket. It is this individual variability we sought to explore further.

**LFPN-DMN connectivity.** LFPN-DMN connectivity was defined as the average correlation of pairs of each ROI in LFPN with each ROI in DMN (each z-transformed; see Methods). Working from our pre-registered analysis plan (https://aspredicted.org/blind.php?x=3d7ry9), we tested the relation between LFPN-DMN connectivity and nonverbal cognitive test performance in the children from households below-poverty. We used linear mixed-effects models to test the association between cognitive test performance and LFPN-DMN connectivity, controlling for children's age and scanner head motion, with a random intercept for study site (see Methods). Contrary to previously published results, we did not find a negative association between LFPN-DMN connectivity and test performance. In fact, the estimated direction of the effect was positive, though this was not statistically

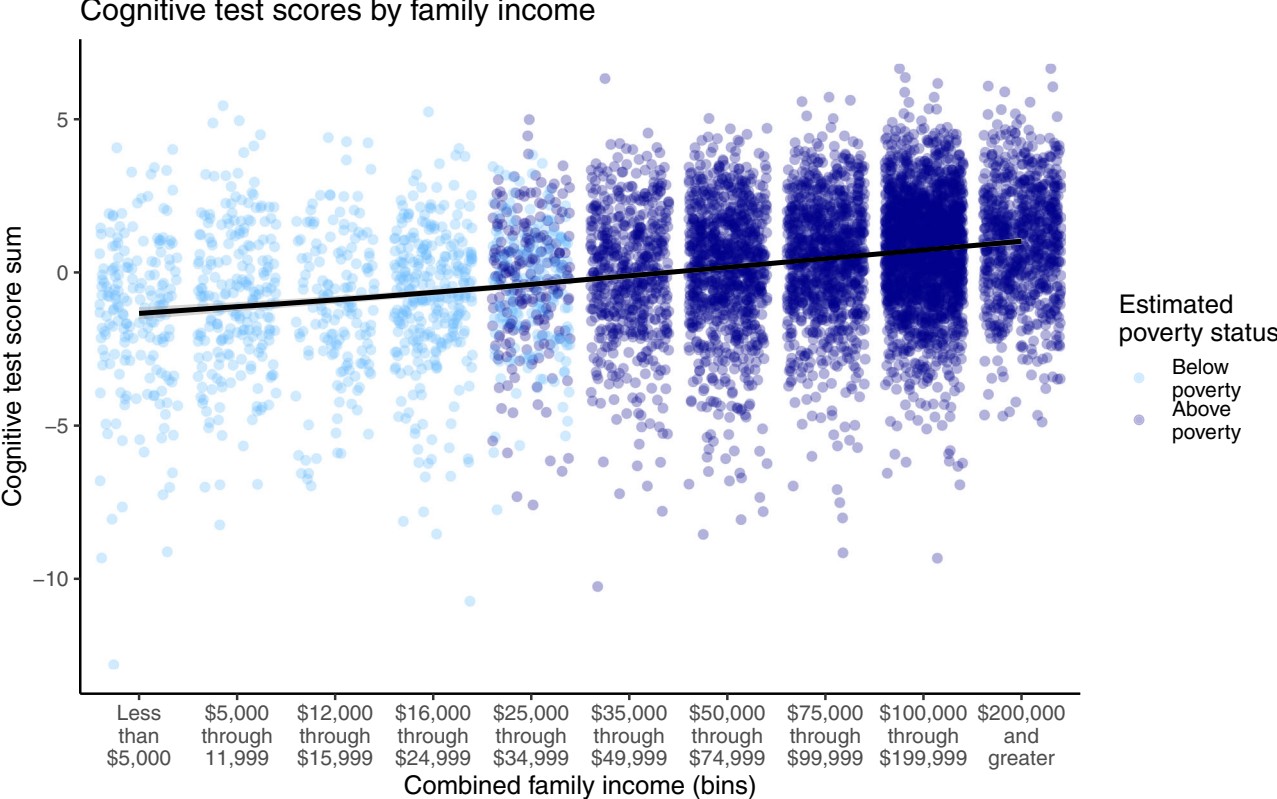

**Fig. 1 Illustration of the variability of cognitive test performance within every level of family income in the sample ($N$ = 6839).** Colors indicate whether children's households were classified as in poverty, based on a combination of their family income and the number of people in the home. Replicating prior studies, higher income is associated with higher cognitive test performance ($R$ = 0.24); however, it is important to acknowledge this substantial variability within and overlap between children at each level of family income.

significant, $B = 2.11$, SE $= 1.12$, $t(1028) = 1.88$; $\chi^2(1) = 3.52$, $p = 0.060$. This numerically positive association was still observed when using a robust linear mixed-effects model, which detects and accounts for outliers or other sources of contamination in the data that may affect model validity, $B = 1.78$, SE $= 1.09$, $t = 1.64$. Thus, this unexpected pattern was not driven by outliers. This effect was most pronounced for Matrix Reasoning and least evident for Flanker, but the estimate was positive for all three tests (see Supplement S4). It was also observed for the NIH Toolbox Fluid Cognition composite score (see Supplement S4).

Given this unexpected result, we next explored whether the expected association between LFPN-DMN connectivity and test performance was present in children from higher-income households in the larger dataset. To this end, we analyzed the 5,805 children from the same study sites who comprised our above-poverty group. Consistent with prior studies[25,34,35], these children showed a negative association between LFPN-DMN connectivity and cognitive test performance, $B = -1.41$, SE $= 0.45$, $t(5794) = -3.14$; $\chi^2(1) = 9.85$, $p = 0.002$. A direct comparison between the samples confirmed that the association between LFPN-DMN connectivity and test performance differed as a function of whether or not children were from households in poverty, $\chi^2(1) = 8.99$, $p = 0.003$ (Fig. 2). For the above-poverty group, having higher LFPN-DMN connectivity appeared to be a risk factor for low cognitive test performance, while for the below-poverty group it tended to be associated with higher performance. Several follow-up tests confirmed the reliability of this dissociation, including a bootstrapping procedure, permutation testing, and tests to ensure that results were not driven by differences in head motion, age, or the specific cognitive measures selected (see Supplement S6–S9).

**LFPN-LFPN connectivity**. LFPN-LFPN connectivity was defined as the average correlation of each ROI pair within LFPN (each z-transformed; see Methods). Following our pre-registration, using linear mixed effects models, we next tested whether children from households in poverty would show the positive correlation between LFPN within-network connectivity and cognitive test performance that has previously been documented. The relation between LFPN-LFPN connectivity and test scores was not significant for the below-poverty group, $B = 0.24$, SE $= 0.87$, $t(1028) = 0.28$; $\chi^2(1) = 0.08$, $p = 0.783$, or for the children from households above-poverty in the larger study, $B = 0.34$, SE $= 0.36$, $t(5797) = 0.94$; $\chi^2(1) = 0.89$, $p = 0.346$. Thus, the strength of resting-state functional connectivity within the LFPN network was not a predictor of cognitive performance in this large sample of 9-to-10-year-olds. As a control for this a priori within-network analysis for LFPN, we conducted an exploratory analysis investigating DMN-DMN connectivity; it exhibited a non-significant interaction with poverty status, $\chi^2(1) = 2.78$, $p = 0.096$.

**Environmental variables**. To further explore the dissociation observed for LFPN-DMN connectivity, we next asked whether features of children's environments might explain why the brain-behavior link differed as a function of poverty status. Even within a particular income group, different children are exposed to very different experiences in their homes, neighborhoods, and schools. We considered 29 demographic variables chosen to reflect features of children's home, school, and neighborhood environments (Supplement S1, 2). To test whether any of these variables could explain the observed group interaction, we performed Ridge regression. Specifically, we used nested cross-validation to predict

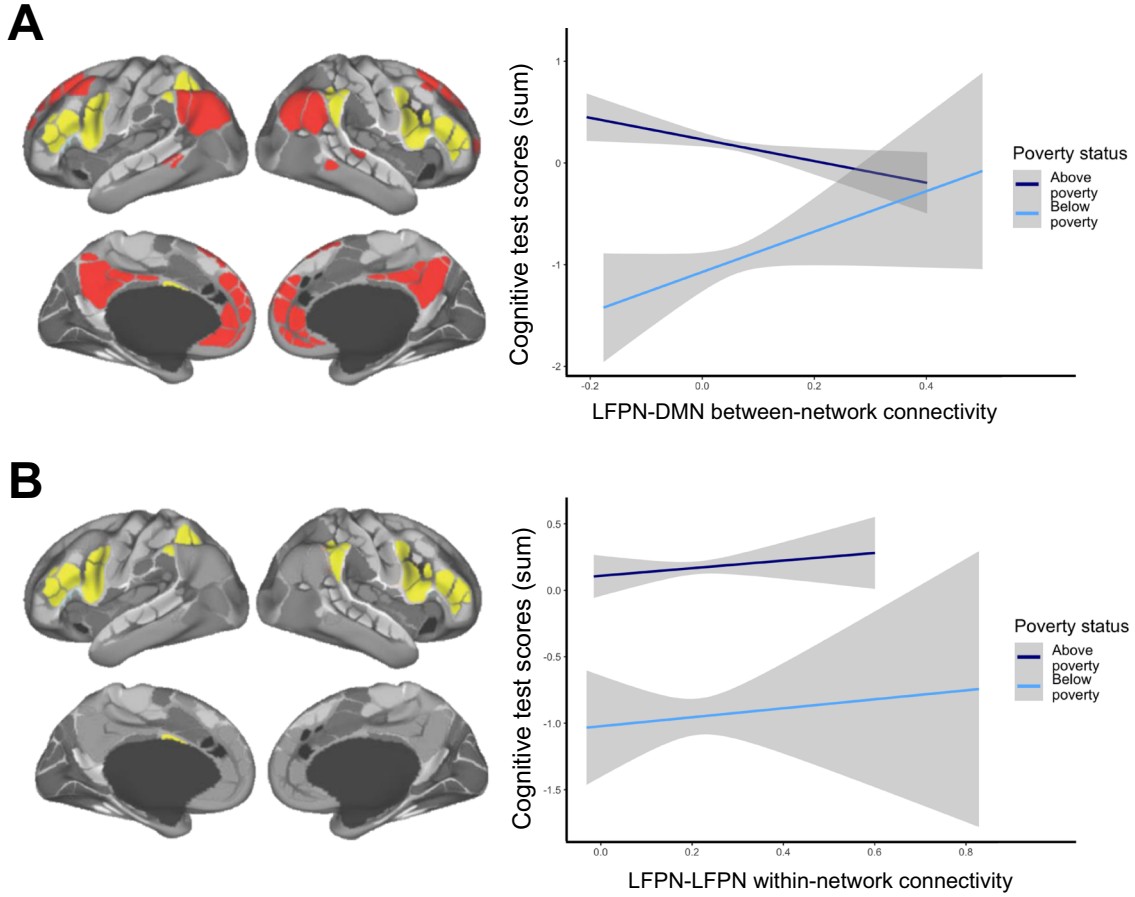

**Fig. 2 Relations between resting state network metrics and cognitive test score residuals, for children living above poverty (dark blue) and below poverty (light blue).** Mixed models include fixed effects for age and motion and a random effect for study site. Data are presented as mean values $+/-95\%$ confidence intervals for a linear model, calculated and displayed using the geom_smooth function in ggplot. **A** Children in households above poverty show an expected, negative, relation between LFPN-DMN connectivity and test performance, $B = -1.41$, $SE = 0.45$; $p = 0.002$, while children in households below poverty show the opposite pattern, $B = 2.11$, $SE = 1.12$; $p = 0.060$, interaction: $X^2(1) = 8.99$, $p = 0.003$. **B** Children across the sample show a non-significant positive relation between LFPN-LFPN within-network connectivity and test performance, above poverty: $B = 0.34$, $SE = 0.36$; $p = 0.346$; below poverty: $B = 0.24$, $SE = 0.87$; $p = 0.783$; interaction: $X^2(1) = 0.0005$, $p = 0.982$. Networks functionally defined using the Gordon parcellation scheme; on left, lateral frontoparietal network (LFPN) is shown in yellow and default mode network (DMN) shown in red, figures adapted from[110] and reprinted by permission from Oxford University Press and the authors.

cognitive test performance from an interaction between LFPN-DMN connectivity and these demographic variables, in addition to the main effects of each of these variables. Briefly, Ridge regression is a regularization technique that penalizes variables that do not contribute to model fit, thus giving more weight to the most important variables. This approach allows for the inclusion of many variables in a model while reducing the chances of overfitting, and deals with issues of multicollinearity. We pre-registered this second step of analyses prior to examining the data further (https://aspredicted.org/blind.php?x=tg4tg9), given the substantial analytic flexibility possible with such a large set of variables.

We trained our model in a training set of two-thirds ($N = 670$, after removing missing data) of the children from households defined as in poverty, using 5-fold cross-validation. Next, we tested whether these demographic and neural model parameters could be used to predict cognitive test scores in the held-out test set: the remaining one-third ($N = 329$) of children from households defined as being in poverty. Indeed, we found that our model performed above chance (cross-validated $R^2_{CV} > 0$; see Supplement S11), explaining 4% of the variance in children's cognitive test scores in this held-out sample. While 4 percent is small, it is on par with the effect of family income on test scores

across the full sample (6%). Additionally, it is a pure indicator, unlike the $R^2$ of models that have been fit to the data themselves and are thus likely to be inflated. Most importantly, this prediction is based on a socioeconomically restricted sample of children: those with a total family income below \$35,000 (below \$25,000 for children in families of 4 or less).

As shown in Table 2, individual, home, neighborhood, and school variables helped to predict cognitive test scores among children living in poverty. Critically, we found that several characteristics of children's experiences interacted with LFPN-DMN connectivity to predict these test scores. Specifically, variables related to school type, neighborhood safety, child's race, and parents' highest level of education contributed to model fit (see Table 2). To better understand these results, we plotted the effects for the factors showing significant interaction effects (Fig. 3). Visualizing the interaction for neighborhood safety revealed that children living in safer neighborhoods, based on parent-reported neighborhood safety, showed a negative relation between LFPN-DMN connectivity and test performance, whereas those who lived in neighborhoods considered less safe based on parent-reported neighborhood safety showed a positive relation. With regard to schooling, the relation between LFPN-DMN connectivity was more positive for children attending public

**Table 2 Estimated coefficients from Ridge regression predicting children's cognitive test scores, when controlling for fixed effects of age and motion and random effects of study site, for all children from households below the poverty line.**

| | Estimate | Scaled estimate | Std. Error (scaled) | t-value (scaled) | Pr(>|t|) |
|---|---|---|---|---|---|
| (Intercept) | 0.12 | NA | NA | NA | NA |
| Black race | −0.10 | −1.46 | 0.28 | 5.29 | 0.000 |
| Parents' highest level of education (years) | 0.05 | 1.53 | 0.32 | 4.76 | 0.000 |
| Census: % of people over age 25 with > = high school diploma | 0.03 | 1.06 | 0.29 | 3.69 | 0.000 |
| White race | 0.06 | 0.98 | 0.29 | 3.42 | 0.001 |
| Asian race | 0.37 | 1.06 | 0.33 | 3.23 | 0.001 |
| Census: % of labor force aged > =16 y unemployed | −0.02 | −0.77 | 0.28 | 2.75 | 0.006 |
| Census: % of families below the poverty level | −0.02 | −0.70 | 0.26 | 2.71 | 0.007 |
| Parent ethnic identification | 0.03 | 0.87 | 0.33 | 2.68 | 0.007 |
| Youth-reported school disengagement | −0.02 | −0.81 | 0.31 | 2.61 | 0.009 |
| Census: income disparity | −0.02 | −0.67 | 0.26 | 2.57 | 0.010 |
| LFPN-DMN x Public school | 0.27 | 0.53 | 0.22 | 2.41 | 0.016 |
| LFPN-DMN x Parent-reported neighborhood safety | −0.19 | −0.67 | 0.29 | 2.35 | 0.019 |
| Census: estimated lead risk | −0.02 | −0.60 | 0.28 | 2.17 | 0.030 |
| LFPN-DMN x Mixed race | 0.74 | 0.65 | 0.31 | 2.07 | 0.038 |
| Third generation American | −0.04 | −0.52 | 0.25 | 2.04 | 0.042 |
| LFPN-DMN x Parents' highest level of education | 0.15 | 0.52 | 0.27 | 1.90 | 0.057 |
| LFPN-DMN | 0.18 | 0.34 | 0.20 | 1.72 | 0.085 |
| LFPN-DMN x Black race | −0.28 | −0.43 | 0.25 | 1.70 | 0.089 |
| LFPN-DMN x non-Hispanic | 0.20 | 0.38 | 0.22 | 1.67 | 0.094 |
| Mixed race | 0.05 | 0.52 | 0.31 | 1.66 | 0.096 |
| LFPN-DMN x White race | 0.31 | 0.46 | 0.28 | 1.61 | 0.107 |
| LFPN-DMN x Not in school | −3.15 | −0.48 | 0.31 | 1.54 | 0.123 |
| LFPN-DMN x Census: % of occupied units without complete plumbing | 0.16 | 0.49 | 0.32 | 1.54 | 0.124 |
| Parent never married | −0.03 | −0.44 | 0.29 | 1.53 | 0.125 |
| First generation American | 0.03 | 0.38 | 0.27 | 1.40 | 0.160 |
| LFPN-DMN x Hours/week spent at another household | −0.14 | −0.46 | 0.33 | 1.39 | 0.165 |
| Second generation American | 0.04 | 0.40 | 0.31 | 1.29 | 0.197 |
| LFPN-DMN x Parent self-reported intrusive behavior | 0.15 | 0.39 | 0.31 | 1.27 | 0.206 |
| Parent-reported neighborhood safety | 0.01 | 0.37 | 0.31 | 1.18 | 0.238 |
| LFPN-DMN x First-generation American | 0.26 | 0.32 | 0.27 | 1.17 | 0.243 |
| LFPN-DMN x Parent ethnic identification | 0.12 | 0.37 | 0.32 | 1.15 | 0.250 |
| Native American/Alaska Native | 0.10 | 0.36 | 0.32 | 1.12 | 0.261 |
| Parent married | 0.02 | 0.33 | 0.30 | 1.11 | 0.266 |
| LFPN-DMN x Census: % of people over age 25 with > = a high school diploma | 0.08 | 0.29 | 0.26 | 1.11 | 0.269 |
| LFPN-DMN x Youth born outside U.S. | 0.83 | 0.36 | 0.33 | 1.09 | 0.274 |
| LFPN-DMN x Private school | −0.70 | −0.35 | 0.32 | 1.09 | 0.278 |
| Other race | −0.04 | −0.33 | 0.31 | 1.07 | 0.286 |
| LFPN-DMN x Parent separated/divorced | 0.25 | 0.31 | 0.29 | 1.06 | 0.288 |
| LFPN-DMN x Youth-reported school involvement | 0.10 | 0.30 | 0.29 | 1.05 | 0.294 |
| LFPN-DMN x Second-generation American | −0.44 | −0.32 | 0.31 | 1.02 | 0.308 |
| Youth-reported parental acceptance | −0.01 | −0.30 | 0.31 | 0.97 | 0.333 |
| Any siblings | −0.02 | −0.30 | 0.33 | 0.90 | 0.366 |
| Other school setting | 0.08 | 0.29 | 0.32 | 0.89 | 0.372 |
| LFPN-DMN x People living in home | −0.06 | −0.27 | 0.31 | 0.87 | 0.387 |
| LFPN-DMN x Third-generation American | 0.10 | 0.19 | 0.23 | 0.86 | 0.392 |
| LFPN-DMN x Youth-reported school disengagement | −0.09 | −0.26 | 0.31 | 0.85 | 0.397 |
| Parent widowed | −0.06 | −0.27 | 0.33 | 0.81 | 0.418 |
| Not in school | −0.11 | −0.25 | 0.31 | 0.80 | 0.425 |
| Home school | −0.16 | −0.22 | 0.30 | 0.73 | 0.463 |
| LFPN-DMN x Financial stress | −0.05 | −0.22 | 0.31 | 0.73 | 0.468 |
| Parent separated/divorced | 0.02 | 0.22 | 0.31 | 0.72 | 0.471 |
| Census: adult violent crime reports | 0.01 | 0.20 | 0.27 | 0.72 | 0.472 |
| LFPN-DMN x home school | −2.82 | −0.21 | 0.30 | 0.71 | 0.478 |
| Youth-reported supportive school environment | −0.01 | −0.21 | 0.30 | 0.70 | 0.483 |
| LFPN-DMN x Asian race | 0.44 | 0.21 | 0.31 | 0.70 | 0.487 |
| LFPN-DMN x Census: income disparity | 0.05 | 0.16 | 0.23 | 0.70 | 0.487 |
| Census: uniform crime reports | 0.01 | 0.19 | 0.28 | 0.68 | 0.498 |
| LFPN-DMN x Youth-reported parental monitoring | −0.06 | −0.21 | 0.31 | 0.67 | 0.503 |
| LFPN-DMN x Any siblings | 0.15 | 0.20 | 0.30 | 0.65 | 0.517 |
| Hours/week spent at another household | −0.01 | −0.21 | 0.34 | 0.63 | 0.526 |
| LFPN-DMN x Native American/Alaska Native | 0.51 | 0.19 | 0.32 | 0.59 | 0.553 |
| LFPN-DMN x Youth-reported family conflict | 0.06 | 0.18 | 0.31 | 0.58 | 0.565 |
| LFPN-DMN x School for behavioral/emotional problems | −2.37 | −0.20 | 0.35 | 0.57 | 0.566 |
| LFPN-DMN x Youth-reported supportive school environment | 0.05 | 0.17 | 0.30 | 0.56 | 0.578 |
| LFPN-DMN x Parent married | 0.11 | 0.16 | 0.28 | 0.55 | 0.580 |
| LFPN-DMN x Census: adult violent crime reports | −0.06 | −0.15 | 0.27 | 0.55 | 0.581 |
| School for behavioral/emotional problems | 0.10 | 0.18 | 0.35 | 0.51 | 0.612 |
| LFPN-DMN x Census: estimated lead risk | 0.04 | 0.13 | 0.25 | 0.50 | 0.616 |
| Youth-reported school involvement | 0.00 | −0.14 | 0.30 | 0.49 | 0.625 |
| People living in home | 0.00 | −0.15 | 0.31 | 0.48 | 0.633 |
| Private school | −0.02 | −0.15 | 0.32 | 0.48 | 0.634 |
| Child born outside U.S. | −0.03 | −0.15 | 0.33 | 0.46 | 0.648 |
| LFPN-DMN x Census: uniform crime reports | −0.05 | −0.13 | 0.28 | 0.45 | 0.650 |
| LFPN-DMN x Other race | −0.17 | −0.13 | 0.31 | 0.44 | 0.661 |
| Youth-reported parental monitoring | 0.00 | −0.13 | 0.32 | 0.42 | 0.671 |
| Parent self-reported aggressive behavior | 0.00 | 0.12 | 0.29 | 0.42 | 0.673 |
| Youth-reported family conflict | 0.00 | −0.12 | 0.32 | 0.39 | 0.695 |
| LFPN-DMN x Charter school | −0.16 | −0.11 | 0.31 | 0.37 | 0.710 |
| Financial stress | 0.00 | 0.11 | 0.33 | 0.35 | 0.726 |
| LFPN-DMN x Head motion | 0.03 | 0.09 | 0.30 | 0.30 | 0.763 |
| LFPN-DMN x Parent never married | 0.05 | 0.07 | 0.27 | 0.26 | 0.795 |
| LFPN-DMN x Parent self-reported withdrawn behavior | 0.02 | 0.08 | 0.30 | 0.25 | 0.802 |
| Head motion | 0.00 | 0.07 | 0.33 | 0.21 | 0.835 |
| LFPN-DMN x Parent self-reported aggressive behavior | 0.02 | 0.06 | 0.29 | 0.19 | 0.847 |
| Hispanic ethnicity | 0.00 | 0.05 | 0.24 | 0.19 | 0.849 |
| Non-hispanic ethnicity | 0.00 | −0.05 | 0.24 | 0.19 | 0.849 |
| Parent self-reported intrusive behavior | 0.00 | 0.06 | 0.31 | 0.19 | 0.852 |
| Age | 0.00 | 0.06 | 0.33 | 0.17 | 0.865 |
| Public school | 0.00 | 0.05 | 0.29 | 0.17 | 0.868 |
| LFPN-DMN x Parent widowed | −0.18 | −0.05 | 0.33 | 0.17 | 0.869 |
| LFPN-DMN x Census: % of families below the poverty level | 0.01 | 0.04 | 0.23 | 0.16 | 0.870 |
| Census: % of occupied units without complete plumbing | 0.00 | 0.05 | 0.33 | 0.16 | 0.873 |
| LFPN-DMN x Youth-reported parental acceptance | 0.01 | 0.04 | 0.30 | 0.13 | 0.900 |
| Parent living with partner | 0.00 | 0.03 | 0.32 | 0.11 | 0.914 |
| LFPN-DMN x Parent living with partner | −0.04 | −0.03 | 0.31 | 0.10 | 0.919 |

**Table 2 (continued)**

| | Estimate | Scaled estimate | Std. Error (scaled) | t-value (scaled) | Pr(>\|t\|) |
|---|---|---|---|---|---|
| *LFPN-DMN x Hispanic ethnicity* | −0.02 | −0.03 | 0.26 | 0.10 | 0.920 |
| *LFPN-DMN x Age* | 0.01 | 0.02 | 0.32 | 0.07 | 0.946 |
| *LFPN-DMN x Other school setting* | 0.03 | 0.01 | 0.32 | 0.03 | 0.976 |
| *LFPN-DMN x Census: % of labor force aged > = 16 y unemployed* | 0.00 | −0.01 | 0.25 | 0.02 | 0.981 |
| Charter school | 0.00 | −0.01 | 0.30 | 0.02 | 0.982 |
| Parent self-reported withdrawn behavior | 0.00 | 0.00 | 0.30 | 0.00 | 0.997 |

Plain text: main effects; *italics*: interactions with and main effect of LFPN-DMN connectivity.
Interactions with and main effect of lateral frontoparietal-default mode network (LFPN-DMN) connectivity italicized.

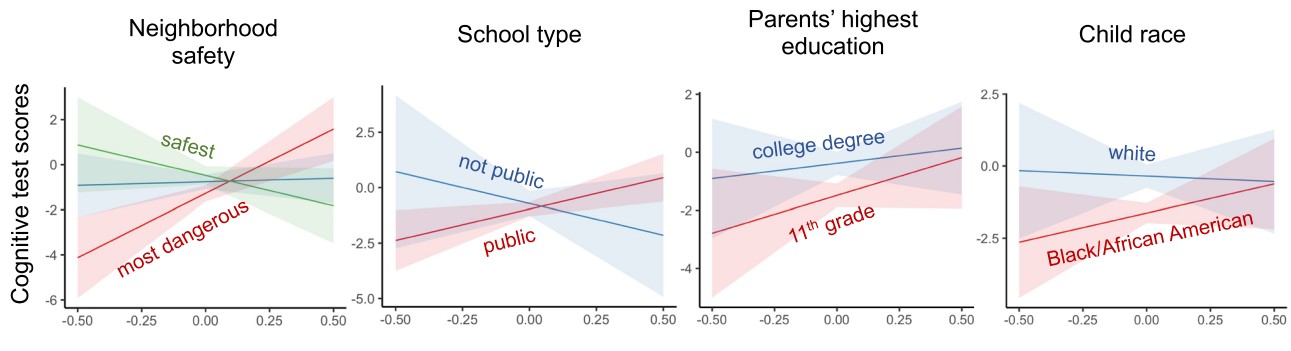

**Fig. 3 Interactions between demographic variables and lateral frontoparietal-default mode network (LFPN-DMN) connectivity in predicting cognitive test scores, for children in households below poverty.** The majority of non-public schools were charter and private schools. In addition, only white and Black/African American race are displayed as these were the most represented in the current sample. Data are presented as mean values +/− 89% level confidence intervals for predicted effects, calculated and displayed using the sjPlot package in R[135].

schools than those attending other types of schools (predominantly charter, $N = 79$, and private, $N = 40$).

Finally, we conducted a confirmatory factor analysis to test whether the demographic variables could be split into individual and home, neighborhood, and school factors based on our a priori categorization. This categorization did not meet our pre-registered criteria for a good model fit (our CFI, 0.11, was considerably lower than 0.9); as a result, we did not continue with this portion of the analysis. Thus, our data-driven approach provided insights that would have been missed by simply categorizing variables based on our prior assumptions about classes of life experiences.

**Exploratory network associations.** Given the differential relation between network connectivity and test performance as a function of the socioeconomic status of the household, we sought to ascertain whether this effect was specific to the LFPN-DMN, or whether there was a more general difference regarding connectivity between networks. Further, we sought to better understand the phenomenon at a conceptual level by assessing the plausibility of several accounts regarding what might constitute adaptive thought patterns for children contending with extremely challenging circumstances. Therefore, we ran several exploratory analyses involving two additional brain networks, selected for the reasons discussed below. Due to the exploratory nature of these analyses, we focus on the general patterns of effects as potentially valuable for guiding future research.

The first additional network in which we tested for effects of poverty status was the cingulo-opercular network (CON), also referred to as the salience network. The CON has been hypothesized to play a role in coordinating the engagement of the LFPN and DMN networks[46,47]. Therefore, we sought to test for differential effects of coordination between the CON and these networks as a function of poverty. We found that weaker

LFPN-CON connectivity was associated with better test performance for both groups, with little evidence of interaction (Fig. 4A). Thus, a dissociation between these networks appears to be generally adaptive at this age. By contrast, DMN-CON connectivity had no main effect on cognitive test performance, but it showed a possible interaction with poverty status (Fig. 4B). Specifically, *weaker* DMN-CON connectivity was directionally associated with better test performance for children from households defined as below poverty, while *stronger* DMN-CON connectivity appeared more adaptive for children from households defined as above poverty. Thus, the DMN is more tightly linked to LFPN and, perhaps, less tightly linked to CON. However, it seems unlikely that a DMN-CON interaction is the key driver of the LFPN-DMN interaction we have uncovered, as the latter effect was stronger. Nonetheless, further research in this population relating these three brain networks to a broader set of cognitive measures is warranted.

The other network we investigated was the retrosplenial temporal network (RTN), which is critical for long-term declarative memory[48,49]. Regions in the RTN interact with the LFPN during performance of episodic memory tasks involving externally-presented stimuli[50,51], but with the DMN during autobiographical memory retrieval[38,52,53] and at rest[54], that is, during internally directed thought. We reasoned that if children from households defined as below poverty that perform well on cognitive tests rely more on their autobiographical memory than do others when facing cognitive challenges, LFPN-RTN connectivity might be positively related to test performance in this sample. Contrary to this prediction, however, we found that *weaker* LFPN-RTN connectivity and DMN-RTN connectivity were associated with better test performance in both the below- and above-poverty samples (Fig. 4C, D). Thus, these exploratory analyses involving the CON and RTN networks may suggest specificity in the observed LFPN-DMN interaction effect.

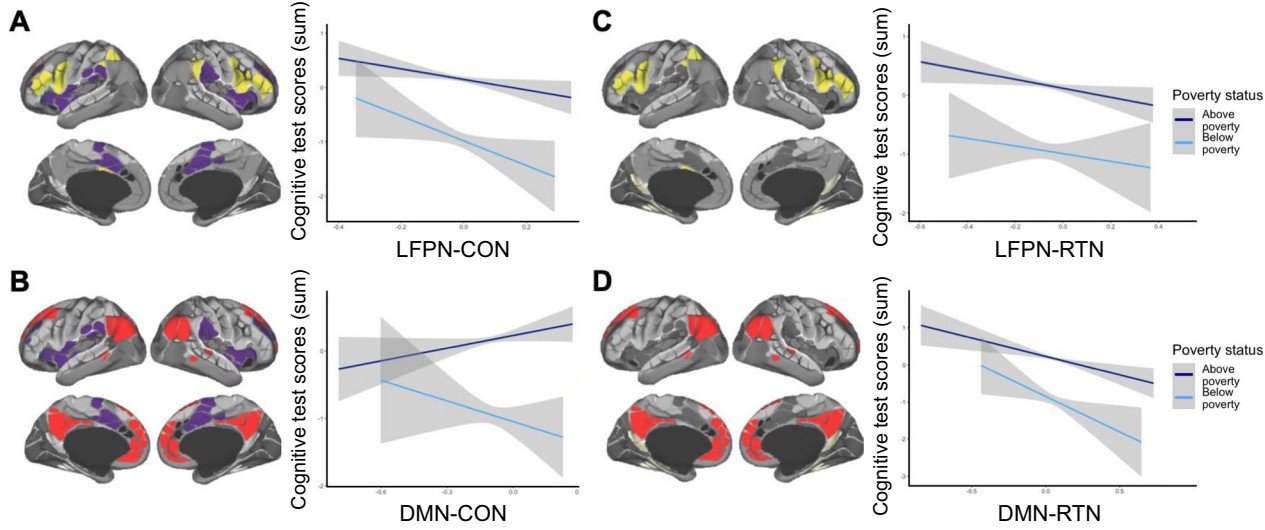

**Fig. 4 Exploratory analyses with the CON (A–B) and RTN (C–D).** As in Fig. 2, plots show relations between resting state network metrics and cognitive test score residuals, for children in households above poverty (dark blue) and below poverty (light blue). Models include fixed effects for age and motion and a random effect for study site. Data are presented as mean values +/− 95% confidence intervals for a linear model, calculated and displayed using the geom_smooth function in ggplot. Networks functionally defined using the Gordon parcellation scheme; lateral frontoparietal network (LFPN) shown in yellow, default mode network (DMN) shown in red, cingulo-opercular network (CON) shown in purple; retrosplenial temporal network (RTN) shown in off-white; figures adapted from[110] and reprinted by permission from Oxford University Press and the authors. **A** Weaker LFPN-CON connectivity was associated with better test performance for both groups, with little evidence of an interaction (main effect: $B = -1.14$, $SE = 0.45$, $t(6824) = -2.53$; $X^2(1) = 11.76$, $p = 0.001$; interaction: $B = -1.42$, $SE = 1.03$, $t(6824) = -1.37$; $X^2(1) = 1.87$, $p = 0.171$). **B** DMN-CON connectivity was not consistently associated with test performance, though it was directionally positive for children in households above poverty and negative for children in households below poverty (main effect: $B = 0.47$, $SE = 0.38$, $t(6823) = 1.24$; $X^2(1) = 0.27$, $p = 0.601$; interaction: $B = -1.66$, $SE = 0.88$, $t(6823) = -1.88$; $X^2(1) = 3.53$, $p = 0.060$). **C**, **D** Weaker LFPN-RTN connectivity and weaker DMN-RTN connectivity were both associated with better test performance, with little evidence of an interaction (**C**: LFPN-RTN main effect: $B = -0.90$, $SE = 0.36$, $t(6829) = -2.54$; $X^2(1) = 7.13$, $p = 0.008$; LFPN-RTN interaction: $B = 0.23$, $SE = 0.84$, $t(6829) = 0.27$; $X^2(1) = 0.08$, $p = 0.784$; **D**: DMN-RTN main effect: $B = -0.99$, $SE = 0.32$, $t(6826) = -3.14$; $X^2(1) = 16.24$, $p < 0.001$; DMN-RTN interaction: $B = -0.95$, $SE = 0.75$, $t(6826) = -1.27$; $X^2(1) = 1.61$, $p = 0.205$).

## Discussion

Prior research in both adults and children suggests that, in order to perform well on cognitively demanding tasks, the LFPN must operate independently from the DMN[33–35], suggesting that it is optimal for individuals engaged in a cognitively demanding task involving externally presented stimuli to focus narrowly on this task while inhibiting internally-directed or self-referential thoughts[23,27,31,32]. However, most of the research that led to this conclusion has been conducted with non-representative samples of individuals from higher-income backgrounds.

In this study, we tested the relation between patterns of brain connectivity and nonverbal cognitive test performance for over 1,000 children in a US sample estimated to be living in poverty. Although these children scored lower on average than their peers in higher-income households from the same study sites, there was large variability. Indeed, many of the children from below-poverty households scored as high as or higher than children whose family incomes were considerably higher. In contrast to prior studies showing a negative correlation between LFPN-DMN connectivity and cognitive performance—that was replicated in our study in the group of children from above-poverty households—our group of children from households below-poverty showed a non-significant *positive* relationship between cognitive performance and functional connectivity between these networks, resulting in significant group interaction.

Confirming the reliability of this dissociation, both a boot-strapping analysis and permutation testing showed that models trained on the data from the children from households above poverty were poor at predicting test performance for the children from households below poverty. Importantly, the most salient difference between children from households below and above

poverty in our analyses was not overall brain connectivity, but rather the relation between connectivity and cognitive performance.

One interpretation of this interaction is that the relation between LFPN-DMN connectivity and test performance depends in part on children's experiences. It may be optimal under some circumstances to engage in thought patterns that more frequently co-activated the LFPN and DMN[37,55,56]. For example, while the DMN is generally thought to be suppressed during goal-directed tasks, it is in fact active during a variety of goal-directed tasks that require internal mentation, or projection outside of the here-and-now[24,38]. We return to this point later in the Discussion.

In contrast to our findings with LFPN-DMN connectivity, we found no significant association between within-network LFPN connectivity and test performance—either in the children in households below or above poverty. These results were unexpected, given prior studies reporting that connectivity within the LFPN is positively related to cognitive test performance in both adults and children[34,57–59]. Of particular relevance to this study, Sherman and colleagues found that for 10-year-olds, higher IQ test performance was correlated with higher connectivity between the dorsolateral prefrontal cortex and the posterior parietal cortex, two hub regions of the LFPN. One reason for the non-significant effect in our study may be that we examined connectivity within the LFPN as a whole, rather than looking at particular regions or subnetworks within LFPN. Thus, the entire network might not be developed enough by ages 9–10 years to see this relationship on a global scale.

To better characterize the positive relation between LFPN-DMN and test performance among the children from households below poverty, we examined a number of environmental variables

on which we expected the group to vary[60,61]. Moreover, experiences that are on average associated with worse cognitive outcomes (such as being deprived of caregiver support in early life) can, under some circumstances, produce *better* cognitive outcomes[62], suggesting there may be different routes to achieving high cognitive performance in these cases. Thus, we predicted that differences in environmental influences *among* children from households below poverty would explain whether strong LFPN-DMN connectivity was adaptive or maladaptive for cognitive test performance.

Our analyses suggested that demographic variables could not be well fit a pre-determined factor structure based on variables relating to the individual, home, neighborhood, and school; therefore, we took a data-driven approach to examine the effects of environmental variables. Because many of these variables are correlated with each other, we adopted an analytic approach—Ridge regression—that allows for collinearity. The results showed that within the population of children from households below poverty, variation in their environments was predictive of their cognitive test performance. We note, however, that this clearly not deterministic; a model trained on two-thirds of the children from households below-poverty explained 4% of the variance in the held-out third, suggesting these variables accounted for a small amount of variance overall.

The most predictive variables in the model may reflect structural barriers that families face, including access to resources and institutions, such as high-quality schools, jobs, and healthcare, stable housing in safe neighborhoods, and experiences of racism within these systems[63–68]. However, the results of this data-driven analysis also raised the possibility that being raised by parents with strong ethnic identification may provide a psychological buffer, in line with other research[69–73].

Notably, we found—in addition to these main effects of demographic variables—several interactions between these variables and LFPN-DMN connectivity that predicted cognitive performance. While Ridge regression precludes us from drawing strong conclusions about the importance of specific variables, we highlight those that contributed significantly to model fit. For example, children from households below-poverty who attended public schools, lived in subjectively more dangerous neighborhoods as assessed by parental reports, and were Black (the next best represented racial group after the white race in our sample below poverty) were more likely to show a positive relation between LFPN-DMN connectivity and test performance.

We considered several possible accounts of the current findings. One possibility is that in order to contend with structural barriers, children experiencing poverty need to monitor their environments (vigilance), as well as their own behavior or performance (self-monitoring), to a greater degree than do other children. This hypothesis stems from research showing that individuals living in poverty are more likely to experience threat in the physical domain (e.g., neighborhood safety[74]) or in the social domain (e.g., racism[65,75]); they are also likely to receive less direct feedback or instruction in crowded or underfunded public schools[76,77] and at home[78]. Additionally or alternatively, children in households in poverty may benefit from thinking more about the past or the future—that is, drawing more on autobiographical memory and future-oriented thinking and planning[38]—or the type of productive mind-wandering that fuels creative insights[37,79,80]. Future research could investigate the possibility that leveraging internally guided cognition is a mechanism of resilience for children from households in poverty. As a first step, one could assess whether children from households in poverty with stronger LFPN-DMN connectivity also show heightened self-monitoring, vigilance, autobiographical memory, and/or creative problem-solving.

Based on the available dataset, we explored the plausibility of these hypotheses by focusing on brain networks that have been associated with monitoring or declarative memory. Specifically, we explored associations of test performance with DMN/LFPN and (1) the CON, to probe whether differences in monitoring and vigilance are likely to play a role; and (2) retrosplenial temporal network (RTN), to assess the plausibility of an account involving autobiographical memory or planning.

While relations with RTN and test performance did not distinguish the children from below- and above-poverty households, we observed a potential interaction between DMN-CON connectivity and poverty status in its association with test performance. Weaker DMN-CON was directionally associated with better test performance for children from households below-poverty, and worse for children from households above poverty. Although this trend-level group interaction involving the CON is unlikely to be the key driver of the LFPN-DMN interaction, it does lend credence to the possibility that monitoring oneself and one's social environment may be one mechanism through which children in households in poverty ultimately score highly on cognitive tests. It is also in line with work suggesting that CON plays a critical role in switching between LFPN and DMN activation[46], that connectivity between the three networks changes across age[81], and that some social cognitive processes rely on all three networks[82].

While our study benefited from the ABCD dataset's rich objective measures of a child's environment, other potential environmental and individual-level variables should be considered in future research[16,83,84]. Future research could also benefit from a more sensitive measure of poverty. Because the publicly available dataset did not specify which of the 19 study sites corresponded to which American city, as this was treated as protected information, we determined a cut-off for our poverty threshold based on cost-of-living across study sites. Because cities across the United States vary substantially in cost-of-living, we selected a stringent cutoff for the poverty line. Thus, there are almost certainly families in the above-poverty group in our study that in practice may be considered below-poverty. In addition, it is important to note that children's performance on cognitive tests can fluctuate from day to day for a variety of reasons[85,86], including motivation[87], that is a likely source of noise in our models.

Further, while we focused on three tests of non-verbal cognitive test performance, future studies should examine a broader range of cognitive systems, as these may be differentially affected by the environment[88]. For example, experiences of threat and deprivation have distinct effects on medial and lateral prefrontal cortex development, respectively;[89] these effects may be mediated in part by lower-level visual and attentional processes[90]. Clearly, there is a need for research that investigates the mechanisms through which the environment affects specific neural and cognitive systems, particularly given that much of this environmental variation is still within the range of experiences that is typical[91]. Overall, these results suggest that different patterns of brain activation for children living in poverty do not imply a deficit[92]. An important next step will be to follow these children longitudinally to see how LFPN-DMN connectivity and its relation with cognitive test performance changes across adolescence.

Another important area of research is to look beyond the canonical cognitive tasks used in the present study to identify assessments or testing contexts for which children living in poverty might be particularly adapted to excel[93]. Doing so might reveal that some children who underperformed on the cognitive measures in the current study have strengths in other domains as a result of adaptation to their environments.

This study opens several questions about the neural underpinnings of these findings that should be further examined. Given

individual variability in network topography[94], future studies should examine whether this variability contributes to our findings and examine alternative parcellation schemes. In addition, LFPN and DMN are both summary network measures; there could be qualitative differences in node-to-node connectivity, or smaller interactions between sub-networks, that we are not capturing in the current study[55,95–97]. Moreover, it would be helpful to look at children's task-based activation and functional connectivity to examine whether children in households in poverty are more likely to activate DMN during neutral, externally driven cognitive tasks outside of their daily environments. Finally, given that these metrics only explain a small amount of variance, it is important to look at the contribution of other neural indices.

Given that the structures that govern academic success have been largely created around the needs of middle- and upper-middle-income families, understanding the strengths of families in poverty—and how children thrive in spite of these structural barriers—is critical[98–101]. Altogether, these results highlight the substantial variability of experiences of children in households in poverty, who are almost always treated as a single, homogenous group in developmental cognitive neuroscience studies and compared to children from higher-SES backgrounds[92]. Moreover, they suggest that our field's assumptions about the generalizability of brain-behavior relations are not necessarily correct. Looking beyond convenience samples of children will ultimately lend more insight into the neural underpinnings of cognition, and may show that there are not universally optimal behavioral or neural profiles. Not only would this advance benefit developmental cognitive neuroscience as a field, but it may ultimately allow us to better serve disadvantaged youth.

## Methods

Analysis plans were pre-registered prior to data access (https://aspredicted.org/blind.php?x=3d7ry9, https://aspredicted.org/blind.php?x=tg4tg9) and analysis scripts are openly available on the Open Science Framework (https://osf.io/hs7cg/?view_only=d2acb721549d4f22b5eeea4ce51195c7). The original data are available with permissions on the NIMH Data Archive (https://nda.nih.gov/abcd). All deviations from the initial analysis plan are fully described in the Supplement S12.

**Participants**. Participants were selected from the larger, ongoing Adolescent Brain Cognitive Development (ABCD) study, which was designed to recruit a cohort of children who closely represented the United States population (http://abcdstudy.org; see[102]). This study was approved by the Institutional Review Board at each study site, with centralized IRB approval from the University of California, San Diego. Informed consent and assent were obtained from all parents and children, respectively. We planned to restrict our primary analyses to children whose households fell below the poverty line on the supplemental poverty measure, which takes into account regional differences in cost-of-living[42]. For example, while the federal poverty level in 2018 was $25,465 for a family of four, the supplemental poverty level in Menlo Park, CA—one of the ABCD study sites—was estimated to be over $37,000 around the same time period. However, upon reviewing the data after our pre-registration, we found that the study site in the ABCD data was de-identified for privacy reasons, and as a result, we could not use study site-specific poverty cut-offs. Instead, we estimated each child's poverty status based on their combined family income bracket, the number of people in their home, and the average supplemental poverty level for the study sites included in the sample.

Based on these factors, we considered children to be in households in poverty if they were part of a family of 4 with a total income of less than $25,000, or a family of 5 or more with a total income of less than $35,000. We made this determination by comparing children's combined household income to the Supplemental Poverty Level for 2015–2017 averaged across study sites[42]. We excluded children who did not provide information about family income and complete data on all three cognitive tests, and/or if their MRI data did not meet ABCD's usability criteria (see below). In addition, due to a scanner error, we excluded post-hoc all children who were scanned on Philips scanners. This left us with 1034 children identified as likely to be living below poverty (6839 across the whole sample). Table 1 provides a breakdown of sample demographics.

**Cognitive test performance**. Children's performance was measured on three non-verbal cognitive tests. Specifically, children completed two tests from the NIH Toolbox (http://www.nihtoolbox.org): Flanker, a measure of inhibitory control[103], and Dimensional Change Card Sort (DCCS), a measure of shifting[104]; and the Matrix Reasoning Task from the Wechsler Intelligence Test for Children-V (WISC-V), a measure of abstract reasoning[105]. Children completed these tests using an iPad synchronized for use with an iPad being controlled by the experimenter[106]. These tests were chosen because they all tax higher-level cognitive skills while having relatively low verbal task demands. We created a composite measure of performance across these three domains by creating z-scores of the raw scores on each of these tests and summing them, as pre-registered; the tests were moderately correlated, $0.23 < r < 0.43$, in the whole sample.

**MRI scan procedure**. Scans were typically completed on the same day as the cognitive battery, but could also be completed at a second testing session. After completing motion compliance training in a simulated scanning environment, participants first completed a structural T1-weighted scan. Next, they completed three to four five-minute resting-state scans, in which they were instructed to lay with their eyes open while viewing a crosshair on the screen. The first two resting-state scans were completed immediately following the T1-weighted scan; children then completed two other structural scans, followed by one or two more resting-state scans, depending on the protocol at each study site. All scans were collected on one of three 3 T scanner platforms with an adult-size head coil. Structural and functional images underwent automated quality control procedures (including detecting excessive movement and poor signal-to-noise ratios) and visual inspection and rating (for structural scans) of images for artifacts or other irregularities;[107] participants were excluded if they did not meet quality control criteria, including at least 12.5 min of data with low head motion during data collection (framewise displacement < 0.2 mm).

**Scan parameters**. Scan parameters were optimized to be compatible across scanner platforms, allowing for maximal comparability across the 19 study sites. All T1-weighted scans were collected in the axial position, with 1mm³ voxel resolution, 256 × 256 matrix, 8 degree flip angle, and 2x parallel imaging. Other scan parameters varied by scanner platform (Siemens: 176 slices, 256 × 256 FOV, 2500 ms TR, 2.88 ms TE, 1060 ms TI; Philips: 225 slices, 256 × 240 FOV, 6.31 ms TR, 2.9 ms TE, 1060 ms TI; GE: 208 slices, 256 × 256 FOV, 2500 ms TR, 2 ms TE, 1060 ms TI). All fMRI scans were collected in the axial position, with 2.4 mm³ voxel resolution, 60 slices, 90 × 90 matrix, 216 × 216 FOV, 800 ms TR, 30 ms TE, 52 degree flip angle, and 6-factor MultiBand Acceleration. The motion was monitored during scan acquisition using real-time procedures to adjust scanning procedures as necessary (see[41]); this prospective motion correction procedure significantly reduces scan artifacts due to head motion[107].

**Resting-state fMRI processing**. Data processing was carried out using the ABCD pipeline and carried out by the ABCD Data Analysis and Informatics Core[107]. T1-weighted images were corrected for gradient nonlinearity distortion and intensity inhomogeneity and rigidly registered to a custom atlas. They were run through FreeSurfer's automated brain segmentation to derive white matter, ventricle, and whole-brain ROIs. Resting-state images were first corrected for head motion, displacement estimated from field map scans, $B_0$ distortions, and gradient non-linearity distortions, and registered to the structural images using mutual information. Initial scan volumes were removed, and each voxel was normalized and demeaned. The signal from estimated motion time courses (including six motion parameters, their derivatives, and their squares), quadratic trends, and meantime courses of white matter, gray matter, and whole brain, plus first derivatives, were regressed out, and frames with greater than 0.2 mm displacement were excluded. While the removal of whole-brain signal (global signal reduction) is controversial in the context of interpreting anti-correlations[108,109], we note that we are able to replicate prior studies showing that a more negative link between our networks of interest is related to test performance in our higher-income sample (see Results), lending credence to the inclusion of this step in the analysis pipeline for our purposes.

The data underwent temporal bandpass filtering (0.009–0.08 Hz). Next, standard ROI-based analyses were adapted to allow for analysis in surface space[107]. Specifically, time courses were projected onto FreeSurfer's cortical surface, upon which 13 functionally-defined networks[110] were mapped and time courses for FreeSurfer's standard cortical and subcortical ROIs extracted[111,112]. Correlations for each pair of ROIs both within and across each of the 13 networks were calculated. These were z-transformed and averaged to calculate within-network connectivity for each network (the average correlation of each ROI pair within the network) and between-network connectivity across all networks (the average correlation of pairs of each ROI in one network with each ROI in another network). Here, we examined only within-network connectivity for LFPN and between-network LFPN-DMN connectivity.

Altogether, the process for curbing potential contamination from head motion was three-fold. First, there was real-time head motion monitoring and correction, as described above, and a thorough and systematic check of scan quality in collaboration with ABCD's Data Analysis and Informatics Center. Second, the signal from motion time courses was regressed out during preprocessing, and frames with greater than 0.2 mm of framewise displacement were excluded from calculations altogether, as were time periods with less than five contiguous low-

motion frames. Third, a final censoring procedure was employed to identify potential lingering effects of motion by excluding any frames with outliers in spatial variation across the brain[107]. In combination, these procedures reduce motion artifacts to the extent possible[113].

**Analysis**. Analyses were performed using R version 3.6.0[114]. We performed two separate linear mixed-effects models using the lme4 package[115] to test the relation between cognitive test scores and (1) LFPN-DMN connectivity, and (2) LFPN within-network connectivity. In our initial pre-registration, we did not consider the nested structure of the data or potential confounds. To determine whether to include these in our model in a data-driven fashion, we tested whether each of the following variables contributed significantly to model fit: (1) nesting within study site, (2) nesting within families, (3) child age, and (4) mean levels of motion in a resting-state scan. All except (2) contributed to model fit at a level of $p < 0.01$ and were thus retained in final models. We note that our reported results are similar when we perform simple linear regression with no covariates, exactly as pre-registered. In addition, results are similar when including all of the covariates in the ABCD study's default LMM package (https://deap.nimhda.org/) – specifically, when adding fixed effects of race, sex, and parent marital status to the same model above. To determine the significance of our neural connectivity metrics, we tested whether these contributed to model fit. In all cases, we compared models without the inclusion of the variable of interest to models with this variable included and calculated whether the variable of interest contributed significantly to model fit, using the ANOVA function for likelihood ratio test model comparison.

In our second set of analyses, we sought to explore the unexpected results from our first set of analyses by asking whether certain environmental variables determine whether LFPN-DMN connectivity is positively or negatively associated with cognitive test performance across individuals. To do this, we gathered 31 environmental variables of interest, spanning home, neighborhood, and school contexts. Upon examining the data, we learned that three of these were not collected at the baseline visit and thus could not be included. Moreover, we made the decision to include ethnicity separate from race, as it was collected, to retain maximal information. The final 29 environmental variables are listed in the Supplement S1. In preparation for our subsequent analyses, we mean-centered and standardized these variables in the larger dataset to allow for potential comparisons across the children from high- and low-income households. Levels of each factor variable were broken down into separate dummy-coded variables for inclusion in factor and ridge analyses. When data were missing, they were interpolated using the mice package in R[116].

We first performed confirmatory factor analysis using the lavaan package in R[117] to see whether individual and home, neighborhood and school variables can be separated into distinct factors. If this achieved adequate fit (significantly better fit than a single-factor model and CFI > 9), we planned to perform a linear mixed-effects model to test the association of cognitive test performance with an interaction between LFPN-DMN connectivity and each factor score.

We next performed a ridge regression using the glmnet package in R[118]. This analysis technique penalizes variables in a model that have little predictive power, shrinking their coefficient closer to zero, thus allowing for the inclusion of many potential predictors while reducing model complexity. These models also include a bias term, reducing the chances of overfitting to peculiarities of the data, a common pitfall of ordinary least squares regression. Finally, ridge regression also deals well with multi-collinearity in independent variables; in contrast to alternatives such as Lasso, if two variables are highly correlated and both predictive of the dependent variable, coefficients of both will be weighted more heavily in a ridge.

We fit ridge regressions predicting cognitive test score residuals, that partialled out the covariates included in our basic linear mixed-effects models (random intercept for study site, fixed effects for age and motion), from an interaction between LFPN-DMN connectivity and each environmental variable of interest. This analysis used nested cross-validation. Specifically, we first split the data into a training (2/3) and testing (1/3) set. We created test score residuals in the training and testing sets separately to avoid data leakage[119], after rescaling the testing data by the training data. We then tuned the parameters of the ridge regression on the training set using 5-fold cross-validation. Ultimately, we used the best-performing model to predict cognitive test scores in the held-out testing set and assessed model fit using $R^2$ cross-validated. An $R^2_{CV}$ above 0 indicates that the model performed above chance; otherwise, it will be below 0. We evaluated the significance of specific variables in our model by plugging in the lambda parameter from the best-performing model to the linear ridge function in the ridge package in R[120], on the whole sample of children from households in poverty.

**Robustness analyses**. We did several additional analyses to test the robustness of our results. First, we repeated our primary analyses as robust linear mixed-effects models, using the robustlmm package in R[121]. These models detect outliers or other sources of contamination in the data that may affect model validity, and perform a de-weighting procedure based on the extent of contamination introduced. Next, we performed a bootstrapping procedure intended to probe how frequently the parameter estimate observed in the children from households below poverty alone would be expected to be observed in a larger population of children from households above poverty (Supplement S6). We also performed a permutation procedure to examine the

extent to which the model parameters from the children from above-poverty households alone could explain the data in the children from households below-poverty (Supplement S7). Finally, given that the children from households below-poverty had significantly more motion than children living above poverty, we repeated our primary analyses with only those children who met an extremely stringent motion threshold of 0.2 mm (Supplement S8), and those who provided the most frames of usable data (Supplement S9).

Additional R packages used for data cleaning, analysis, and visualization include: dplyr[122]; ggplot2[123]; car[124]; corrplot[125]; MuMIn[126]; tidyr[127]; summarytools[128]; finalfit[129]; fastDummies[130]; caret[131]; scales[132]; foreign[133]; MASS[134]; sjPlot[135]; tableone[136]; gtools[137].

**Reporting Summary**. Further information on research design is available in the Nature Research Reporting Summary linked to this article.

## Data availability
All raw and processed data used for these analyses are available in the ABCD Data Repository in the National Institute of Mental Health (NIMH) Data Archive Collection #2573 (https://nda.nih.gov/abcd). To obtain permission to these data, users must create an account through the NIMH Data Archive and follow the instructions on the website to gain access.

## Code availability
All analysis scripts used for the current study are publicly available on the Open Science Framework (https://osf.io/hs7cg/?view_only=d2acb721549d4f22b5eeea4ce51195c7).

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

## Acknowledgements

This study would not be possible without the massive efforts of the large team of ABCD leaders and organizers, staff and data curators, and families and children who participated. Research reported in this publication also benefited from the ABCD Workshop on Brain Development and Mental Health, supported by the National Institute of Mental Health of the National Institutes of Health under Award Number R25MH120869. We are grateful to Mahesh Srinivasan for his thoughtful comments on a previous draft of this manuscript, and to the members of the Building Blocks of Cognition Lab and the Language and Cognitive Development Lab for their feedback along the way. The content is solely the responsibility of the authors and does not necessarily represent the official views of the National Institutes of Health. MEL was supported by NSF GRFP DGE 1752814. SAB was supported by a Jacobs Foundation Advanced Career Research Fellowship.

## Author contributions

M.E.L., S.W.G., and S.A.B. conceptualized the study. M.E.L. analyzed the data under the guidance of S.A.B. M.E.L. and S.A.B. wrote the paper, and S.W.G. reviewed the paper.

## Competing interests

The authors declare no competing interests.
