## [Peer Review File · Nature Communications]

Brain network coupling associated with cognitive performance varies as a function of a child's environment in the ABCD studyEditorial Note: This manuscript has been previously reviewed at another journal that is not operating a transparent peer review scheme. This document only contains reviewer comments and rebuttal letters for versions considered at *Nature Communications*. Mentions of the other journal have been redacted.

Reviewer #1 (Remarks to the Author):

I got the chance to review this manuscript [redacted], and feel that the revised version has done a great job addressing the comments I had on the original submission.

I do still have 2 remaining comments -- 1 minor and 1 a bit more significant

1. The title should allude to the fact that the authors took a focused view of FPN-DMN FC, rather than "network coupling"—which seems to imply a wholebrain effect. Relatedly, given the focus on FPN and DMN, it seems odd that the authors didn't also consider within DMN FC. I believe there are several studies linking reduced intra-DMN FC to developmental psychopathology. Further, given the tripartite model of developmental psychopathology I would have also preferred it if the authors also considered SN—but I get that they preregistered the methods so leaving SN out is fine.

2. What is the range of proportion of censored volumes (>0.2mm FD) across the sample? Given studies demonstrating increased reliability of FC as a $f(x)$ of scan length, it would be important to ensure that the observed interaction effect was not driven by differences in the proportion of good data across kiddos below versus above poverty (seems likely given mean FD differences between groups). One way to do this might be to plot the difference between LFPC-DMN FC between the full scan and subsets of the scan at various lengths (cf. Figure 2A from Gordon et al, 2017, *Neuron*). If >12.5 mins is enough for their targeted FC estimate to stabilize, then I agree that the observed effect is unlikely to be contaminated by motion. But as it stands, covarying for mean FD might not be enough if the groups also differ in the amount of data used to generate their FC estimates.

Overall, I think this is a solid contribution to the field.

Take care,
Jeremy Hogeveen, PhD

Reviewer #2 (Remarks to the Author):

The authors have addressed my comments.

Reviewer #3 (Remarks to the Author):

I have reviewed the manuscript "What is an adaptive pattern of brain network coupling for a child? It depends on their environment" before for another journal. With this new version, the authors have addressed my points and improved the manuscript. I have two further minor suggestions below:

1. Abstract: the authors may want to consider switching the order of the two last sentences. One could connect the pieces to "This significant interaction related to several features of a child's environment, suggesting that "optimal" brain function depends in part on the external pressures children face. Future research should investigate the possibility that leveraging internally guided cognition is a mechanism of resilience for children in poverty. Further, our study highlights the need

for more research on more diverse samples in research on the human brain and behavior.
2. Line 371: instead of “reveal” I would be more cautious and say “may suggest” since the authors have not explicitly tested for specificity in the whole brain network.

Reviewer #4 (Remarks to the Author):

Thank you for the opportunity to review “What is an adaptive pattern of brain network coupling for a child? It depends on their environment.”

Overall I think this is a well-written and very well-thought out study. I commend the authors for a) focusing on adaptations to the environment and not taking a deficit-based approach b) preregistering their hypotheses and c) making their code and data publicly available. I also believe the authors did a thorough and compelling job addressing the concerns of the previous reviewers (I did not review the original submission of this paper).

My one remaining concern lies in the robustness of the results based on the selection of regions of interest to define the networks of interest (in particular the main analyses of the LFPN and DMN). Emerging evidence suggests that network parcellation choice can impact the results of studies looking at individual differences in average network connectivity. In a recent study I saw at a conference presentation (currently unpublished) Bryce and colleagues (Flux Congress, 2020) compared within network connectivity in several networks defined by several different average parcellations (including Gordon 2016; Yeo et al., 2011; Power et al., 2011; and Glasser et al., 2016). In looking at the association between average within-network connectivity and individual differences (separate tests for age, poverty, and cognitive performance), and find that the significance of the association depends on which parcellation is chosen. Thus, I suggest that the authors rerun at least their main analyses (LFPN-DMN, and LFPN-LFPN) using these other network parcellations to confirm the robustness of the these findings.

I think this paper makes an important contribution to the field. But I think it is prudent to confirm these results with different network parcellations.

Response to Reviewer Comments

Reviewer #1 (Remarks to the Author):

I got the chance to review this manuscript [redacted], and feel that the revised version has done a great job addressing the comments I had on the original submission.

I do still have 2 remaining comments -- 1 minor and 1 a bit more significant

1. The title should allude to the fact that the authors took a focused view of FPN-DMN FC, rather than "network coupling"—which seems to imply a wholebrain effect. Relatedly, given the focus on FPN and DMN, it seems odd that the authors didn't also consider within DMN FC. I believe there are several studies linking reduced intra-DMN FC to developmental psychopathology. Further, given the tripartite model of developmental psychopathology I would have also preferred it if the authors also considered SN—but I get that they preregistered the methods so leaving SN out is fine.

We appreciate that the title (which has now been changed for other reasons) isn't perfectly precise, but think it is more feasible for us to specify the specific networks we focus on in the abstract, given the general audience of the journal.

We have now tested DMN within-network connectivity, and added this to the main text:

As a control for this a priori within-network analysis for LFPN, we conducted an exploratory analysis investigating DMN-DMN connectivity; it exhibited a non-significant interaction with poverty status, $\chi^2(1) = 2.78, p = 0.096$.

We would like to clarify that we did in fact examine SN connectivity; we refer to it as the cingulo-opercular network (CON) rather than the salience network. We describe results on page 12 and discuss them on page 16-17. We previously indicated in the discussion section that this network is sometimes referred to as the salience network; however, we now flag this from the outset on page 12.

2. What is the range of proportion of censored volumes (>0.2mm FD) across the sample? Given studies demonstrating increased reliability of FC as a f(x) of scan length, it would be important to ensure that the observed interaction effect was not driven by differences in the proportion of good data across kiddos below versus above poverty (seems likely given mean FD differences between groups). One way to do this might be to plot the difference between LFPC-DMN FC between the full scan and subsets of the scan at various lengths (cf. Figure 2A from Gordon et al, 2017, Neuron). If >12.5 mins is enough for their targeted FC

estimate to stabilize, then I agree that the observed effect is unlikely to be contaminated by motion. But as it stands, covarying for mean FD might not be enough if the groups also differ in the amount of data used to generate their FC estimates.

Scan length is a relevant metric that we had not previously considered in our analyses. As we now report, our finding of an interaction between LFPN-DMN connectivity and poverty status still holds when considering scan length. We include in our supplement both an analysis that covaries for scan length, and analyses with the top quartiles of usable data (e.g., those who have the longest scan lengths). We show that our findings are consistent across these various analysis specifications, even for subsets of participants with the most data.

We have added the following to the Supplement (p. 6-7):

Relations between LFPN-DMN connectivity and cognitive test performance, controlling for number of usable frames

A related concern is that our finding was driven by group differences in the number of usable frames of resting state data. Indeed, resting state metrics become more stable with more data (Gordon et al., 2017). In our data, the number of frames participants contributed after outliers were excluded ranged from 376-2170. We also found that LFPN-DMN connectivity was related to participants' number of usable frames, even when controlling for mean framewise displacement, $\chi^2(1) = 21.23$, $p < 0.001$. However, frames of usable data no longer contributed to model fit when considering participants with relatively more usable frames (top 75% of usable frames, >759 : $\chi^2(1) = 2.03$, $p = 0.154$; top 50% of usable frames, >1005 : $\chi^2(1) = 1.34$, $p = 0.247$; top 25% of usable frames, >1199 : $\chi^2(1) = 1.36$, $p = 0.244$).

To address whether scan length affected our results, we first reran our primary model testing the interaction between LFPN-DMN and poverty status in predicting cognitive test scores, with the additional covariate of number of usable frames after outliers were removed. The interaction between LFPN-DMN and poverty status remained significant, $B = 3.14$, $SE = 1.06$, $t(6825) = 2.97$, $\chi^2(1) = 8.81$, $p = 0.003$, and the number of usable frames did not contribute to model fit above and beyond framewise displacement, $\chi^2(1) = 1.91$, $p = 0.167$. (Framewise displacement continued to contribute significantly to model fit, $\chi^2(1) = 20.43$, $p < 0.001$.) Moreover, the interactive effect remained when restricting analyses to only those participants in the top 75th, 50th and 25th percentiles of usable frames (see Supplementary Figure 4 below for results and associated Ns).

Supplementary Figure 4.

Interactions between LFPN-DMN connectivity and poverty status (children estimated to be living above poverty, dark blue, and below poverty, light blue) in predicting cognitive test performance. Left: children with greater than 759 frames after outliers have been excluded, representing 75% of the sample; center: children with greater than 1005 frames after outliers have been excluded, representing 50% of the sample; right: children with greater than 1199 frames after outliers have been excluded, representing 25% of the sample.

Overall, I think this is a solid contribution to the field.

Thank you for your kind words.

Take care,
Jeremy Hogeveen, PhD

Reviewer #2 (Remarks to the Author):

The authors have addressed my comments.

Reviewer #3 (Remarks to the Author):

I have reviewed the manuscript "What is an adaptive pattern of brain network coupling for a child? It depends on their environment" before for another journal. With this new version, the authors have addressed my points and improved the manuscript. I have two further minor suggestions below:

1. Abstract: the authors may want to consider switching the order of the two last sentences. One could connect the pieces to "This significant interaction related to several features of a child's environment, suggesting that "optimal" brain function depends in part on the external pressures children face. Future research should investigate the possibility that leveraging internally guided cognition is a mechanism of resilience for children in poverty. Further, our study highlights the

need for more research on more diverse samples in research on the human brain and behavior.

We have now taken out the sentence about future research in line with the editorial comments from Nature Communications; however, we have incorporated the other suggestion from the reviewer. This part of the abstract now reads:

This significant interaction related to several features of a child's environment, suggesting that what could be considered a beneficial pattern of brain function depends in part on the external pressures children face. These results highlight the importance of studying diverse sample populations.

2. Line 371: instead of “reveal” I would be more cautious and say “may suggest” since the authors have not explicitly tested for specificity in the whole brain network.

Thank you. This sentence now reads: “Thus, these exploratory analyses involving the CON and RTN networks may suggest specificity in the observed LFPN-DMN interaction effect.” (page 14).

Reviewer #4 (Remarks to the Author):

Thank you for the opportunity to review “What is an adaptive pattern of brain network coupling for a child? It depends on their environment.”

Overall I think this is a well-written and very well-thought out study. I commend the authors for a) focusing on adaptations to the environment and not taking a deficit-based approach b) preregistering their hypotheses and c) making their code and data publicly available. I also believe the authors did a thorough and compelling job addressing the concerns of the previous reviewers (I did not review the original submission of this paper).

Thank you for the kind words about our study.

My one remaining concern lies in the robustness of the results based on the selection of regions of interest to define the networks of interest (in particular the main analyses of the LFPN and DMN). Emerging evidence suggests that network parcellation choice can impact the results of studies looking at individual differences in average network connectivity. In a recent study I saw at a conference presentation (currently unpublished) Bryce and colleagues (Flux Congress, 2020) compared within network connectivity in several networks defined by several different average parcellations (including Gordon 2016; Yeo et al., 2011; Power et al., 2011; and Glasser et al., 2016). In looking at the association between average within-network connectivity and individual differences (separate tests for age, poverty, and cognitive performance), and find that the significance of the association depends on which parcellation is chosen. Thus, I suggest that

the authors rerun at least their main analyses (LFPN-DMN, and LFPN-LFPN) using these other network parcellations to confirm the robustness of the these findings.

I think this paper makes an important contribution to the field. But I think it is prudent to confirm these results with different network parcellations.

We have looked into the possibility of rerunning our analyses with other network parcellations. To do so, however, we would likely need to wait until mid or late fall, when the ABCD team releases additional data to the public. We have spoken to a member of the ABCD team, who estimated that the next release will happen *around* September. Even under this best-case scenario, we would need time to conduct these additional analyses and write them up, and then get the manuscript rereviewed. We are concerned that delaying publication by another ~6 months could reduce the novelty of our study if a similar finding were to be published in the interim.

More to the point, perhaps, it is not clear to us how much we would gain by conducting these re-analyses. If we were to measure average connectivity between different expanses of cortex than what we are examining here, it wouldn't necessarily be surprising or concerning if we don't get the same results -- not any more than if we were to find a difference in average level of activity between partially non-overlapping pools of neurons.

We have, however, added to the discussion the point that it may be of interest in the future to test whether these results are obtained with different cortical brain parcellations.

Reviewer #1 (Remarks to the Author):

No additional concerns -- Strong response and the revised manuscript looks ready to me!

Reviewer #2 (Remarks to the Author):

I think the authors have done a good jobs on the paper.

Reviewer #3 (Remarks to the Author):

The authors have addressed my comments.

Reviewer #4 (Remarks to the Author):

Thank you for the opportunity to review the revision of the manuscript now titled "Brain network coupling associated with strong cognitive performance depends on a child's environment." The authors have done an excellent job addressing the concerns of the reviewers. I appreciate the authors looking into the possibility of re-analyzing the data using different parcellations but that this would be excessively cumbersome and delay publication of an important manuscript. I think this manuscript makes an important contribution to the field and I have no further comments.